# Evaluation of Therapeutic Capability of Emustil Drops against Tear Film Complications under Dry Environmental Conditions in Healthy Individuals

**DOI:** 10.3390/medicina59071298

**Published:** 2023-07-14

**Authors:** Ali Abusharha, Ian E. Pearce, Tayyaba Afsar, Suhail Razak

**Affiliations:** 1Department of Optometry, College of Applied Medical Sciences, King Saud University, Riyadh 11433, Saudi Arabia; aabusharha@ksu.edu.sa; 2Vision Sciences, Glasgow Caledonian University, Glasgow G4 0BA, UK; e.i.pearce@gcu.ac.uk; 3Department of Community Health Sciences, College of Applied Medical Sciences, King Saud University, Riyadh 11433, Saudi Arabia

**Keywords:** dry eye disease (DED), tear evaporation rate, Emustil, controlled environment chamber, TearScope Plus, environmental stress

## Abstract

*Background and Objectives*: Dry eye disease (DED) is a multifactorial ailment of the tears and ocular surface. The purpose of this study was to assess the tear film physiology under controlled dry environmental conditions and compare the efficacy of oil-in-water emulsion drops on tear film parameters in protection and relief treatment modalities under low-humidity conditions. Emustil eye drops were used after exposure to a low-humidity environment in the relief method, whereas, in the protection method, the drops were applied before exposure to low humidity. *Materials and Methods*: 12 normal male subjects (mean age 34.0 ± 7.0 years) were exposed to ultra-dry environmental conditions. A number of tear film measurements were carried out under desiccating environmental conditions in a controlled environment chamber (CEC), where the chamber temperature sat at 21 °C with a relative humidity (RH) of 5%. Keeler’s TearScope Plus and an HIRCAL grid were used to assess the tear break-up time and lipid layer thickness (LLT), and the evaporation rate was evaluated using a Servomed EP3 Evaporimeter. *Results*: LLT measurements showed that the dry environment affected LLT significantly (*p* = 0.031). The median grade of LLT dropped from grade 3 (50–70 nm) at 40% RH to grade 2 (13–50 nm) at 5% RH. A significant increase in LLT was seen after both modes of treatment, with a median LLT grade of 3 when the Emustil was used for both protection (*p* = 0.004) and relief (*p* = 0.016). The mean tear evaporation rate in normal environmental conditions (40%) was 40.46 ± 11.80 g/m^2^/h (0.11 µL/min) and increased sharply to 83.77 ± 20.37 g/m^2^/h (0.25 µL/min) after exposure to the dry environment. A minimal decrease in tear film evaporation rate was seen in relief; however, statistical tests showed that the decrease in tear film evaporation rate was not significant. Mean NITBUT dropped from 13.6 s at 40% RH to 6.6 s at 5% RH *(p* = 0.002). All NITBUT measurements at 5% RH (with or without the instillation of Emustil) were significantly lower than those at 40%. The instillation of Emustil at 5% RH resulted in a significant improvement in NITBUT for protection (*p* = 0.016) but this was not the case for relief (*p* = 0.0.56). *Conclusions:* A control environmental chamber (CEC) enables the analysis of tear film parameters comparable to those found in dry eye patients. This enables us to examine the capability of oil in emulsion drops to manage tear film disruption in healthy individuals. This study suggests that using Emustil oil-in-water emulsion before exposure to a dry environment should be advocated for people who work in dry environments.

## 1. Introduction

Dry eye disease (DED) is a multifactorial ailment that causes visual blurriness, irritation, tear film instability and compromised visual quality. DED can be categorized into two primary subtypes [1]: aqueous insufficiency dry eye (due to reduced aqueous secretion from lacrimal glands) and evaporative dry eye (due to a deficient lipid layer). DED can develop as a result of various intrinsic factors, such as ageing; hormonal imbalance [2]; systemic or local autoimmune illness; diabetes; vitamin A deficiency; dehydration; thyroid dysfunction; etc. [3,4], or extrinsic factors, i.e., exposure to adverse environmental conditions (including humidity, temperature, airflow or air pollution); contact lens wear; or refractive surgery [5,6].

The lipid layer is an essential component of the tear film, providing a uniform optical surface for the cornea and maintaining the tear film homeostasis and evaporation rate [7]. The tear film itself exhibits a higher viscosity at the lower temperature of the ocular surface. The meibomian lipids maintain the tear film by dropping its free energy; they transport water into the film during its formation and work together with lipid-binding proteins in the aqueous phase [6]. Tear film lipid thickness is strongly affected by exposure to a dry environment [7], as thinning and changes in tear film lipid patterns have been found after exposure to a relative humidity of 5% for 15 min [8].

Presently, a considerable amount of the population is exposed to adverse atmospheric conditions or artificial environments [9]. There is a link between the rise in temperature and low relative humidity (RH) with the thinning of the tear film, a reduction in tear break-up time, and an intensification of dryness in eyes [10]. The treatment for advanced complications involves eye drops containing anti-inflammatory corticosteroids and antibiotics in addition to omega 3 fatty acid or tear secretagogues [11]. In order to mimic the composition of tear film, which is also water-based, oil-in-water emulsion formulations (containing both water and lipid) have been introduced as novel tools in ophthalmic medication. Lipid-based emulsions (Les) have great potential for the delivery of hydrophobic drugs because of an enhanced corneal permeability, less toxicity and high retention time. Oil-in-water emulsions are composed of small droplets of oil that are stabilized by emulsifiers or surfactants and distributed throughout an aqueous medium. Different drugs like azithromycin, difluprednate, etc., have been delivered by this type of LE. Numerous lipid-based therapies have been employed in the current scenario. Several types of oils (e.g., soybean oil) and surfactants (e.g., Tween 80) are used as intimate components of Les. The choice and selection of the components are based on the parameters, such as the toxicity, irritation potential, and mechanism of action [12]. 

Emustil^®^ (Moorfield Pharmaceutical, London, UK) is a lubricating eye drop, also known as an ‘artificial tear’. Emustil^®^ eye drops contain two ingredients—7% soybean oil and 3% natural phospholipids derived from egg yolk. Soybean oil relieves eye dryness and soreness. Natural phospholipids are substances that are naturally found in tears, but that can be reduced in people with dry eyes [13]. Emustil contains both polar and non-polar lipids and is designed to enhance and thicken the tear film lipid layer [14]. The role of oil–water emulsion in improving subjective symptoms has been studied previously. Emulsion drops have been found to significantly improve tear film parameters such as the evaporation rate, non-invasive tear break-up time (NITBUT) and lipid layer thickness in dry eye patients. Studies have shown an increase in tear film lipid layer thickness after the instillation of oil–water emulsion [15,16]. A recent study compared three over-the-counter tear supplements and found that an oil-in-water (Emustil) in addition to sodium hyaluronate and hydroxypropyl methylcellulose significantly improved symptoms and tear film evaporation in dry eye patients [17]. The same report also found that tear osmolarity and corneal staining were significantly improved among dry eye patients after using Emustil oil-in-water emulsion [17]. A significant increase in meibomian lipid secretion as the eyelid temperature is increased has been noted [18].

Various investigations on testing the efficacy of oil-in-water emulsions have been carried out under normal environmental conditions. As yet, there has been no study conducted to investigate the immediate acute effect of oil-in-water when used under adverse conditions in normal healthy subjects or dry eye patients. The purpose of this study was to assess the tear film physiology under controlled dry environmental conditions and compare the efficacy of oil-in-water emulsion drops on tear film parameters in protection and relief treatment modalities under low-humidity situations. Therefore, we examined the efficiency of an oil-in-water formulation used to protect (installed pre-exposure to low RH) and relieve (installed post-exposure to low RH) the changes in tear film parameters that are caused by exposure to ultra-dry environmental conditions (5% RH). In addition, the study aimed to evaluate the efficacy of using tear supplements during pre-exposure and post-exposure to dry conditions in order to determine the most efficient approach. We employed a control environmental chamber (CEC) to create conditions of environmental stress for exposing the normal subjects to investigate tear film parameters comparable to those found in dry eye patients. This enables us to examine the capability of novel tear supplements to manage tear film disruption in healthy individuals.

## 2. Methodology

### 2.1. Participants

This non-randomized, observational and comparative study analyzed the tear film parameters of 12 normal male subjects (mean age 34.0 ± 7.0 years). The study was approved by the Institutional Review Board, Glasgow Caledonian University Ethics Committee. All participants provided written and oral informed consent, which included details about the study procedures and instructions.

Subjects were first invited to attend a screening visit where tear stability, lipid layer and evaporation rate were assessed. An HIRCAL grid [19] was utilized to assess tear film stability. Subjects were enrolled in this study if they had a Schirmer strip wetting length of more than 10 mm in 5 min, a non-invasive tear break-up time of more than 10 s and ocular symptoms score (Ocular Surface Disease Index OSDI) of less than 12.

### 2.2. Control Environmental Chamber (CEC)

Relative humidity (RH) and ambient temperature were programmed using a controlled environmental chamber (CEC). It is an isolation room measuring 3 × 3 × 2 m and was designed and built by Weiss-Gallenkmap (Loughborough, UK). This controlled environment chamber (CEC) has the capacity to generate temperatures ranging from 5 to 35 °C (±2 °C) with relative humidity levels between 5 and 95% (±3%). The chamber’s humidity regulator automatically controls the relative humidity by activating or deactivating the humidification system and adjusting the power to the dehumidification unit.

Two environmental conditions were created in the CEC:Normal environment 40% RH at 21 °C.Desiccating environment 5% RH at 21 °C.

### 2.3. Study Design

Emustil contains both polar and non-polar lipids and is made from 7% soybean oil and 3% natural phospholipids derived from egg yolk. This study aimed to assess the efficiency of Emustil formulation (Moorfield Pharmaceuticals, London, UK) in protection (before exposure) or relief (post-exposure) on the human tear film parameters in adverse dry conditions.

The tear film parameters of the subjects who met the inclusion criteria were assessed during the screening visit under normal environmental conditions (40% RH/21 °C). The subjects were assigned randomly into two groups to measure the tear parameters under dry conditions using two different methods, protection and relief. The tear film parameters were observed 15 min following exposure to 5% RH before the use of Emustil. Then, the drop was instilled into the eye after 15 min of exposure to the dry environment to determine if any relief was experienced by the subjects (relief method). In the protection method, Emustil tear supplement was administrated 15 min prior to the exposure to dry environmental condition (Figure 1).

### 2.4. Parameter Assessed

A Servo-Med Evapometer was used to measure the tear evaporation rate while the thickness of tear lipid layer and tear break-up time was assessed using Keeler’s Tearscope Plus [20,21].

### 2.5. Measurement of Lipid Layer Thickness (LLT)

This was performed using a Tearscope Plus with a non-illuminated biomicroscope that provided a cold-cathode wide angle lighting system (Keeler Ltd., Berkshire, UK) [21]. The participants were positioned at a slit lamp and instructed to blink naturally while maintaining their gaze as fixed on the light source directly. A sharp and clear image of the interferometric patterns was then obtained by focusing the system carefully. Then, the interferometric patterns were classified according to the grading system of Guillon and Guillon [21].

### 2.6. Tear Film Evaporation Rate (TFER)

In order to measure tear film evaporation rate of the tear film, a modified Servo-Med Evapometer was used [20]. The evaporation rate from the ocular surface can be estimated by calculating the vapor pressure gradient between two points separated by a known distance using Fick’s law:$$\frac{1}{A}\times \frac{dm}{dt}=-D^{\prime }\times \frac{dp}{dx}$$
where (1/*A*dm*/*dt*) is the evaporation rate (g/m^2^/h), *D*′ is a constant (0.607 g/m^2^/h Pa) and *dp*/*dx* is the vapor pressure gradient in the air layer adjacent to the evaporating surface.

A probe attached to a slit lamp was equipped with a pair of sensors, one for temperature and the other for humidity. A swimming goggle was utilized to attach the probe at a specific distance from the eye in order to isolate the ocular surface from air flows and the surrounding environment. Goggle was maintained firmly by the subjects over the right eye to prevent any gap between the goggle and the skin surface. Each subject underwent two measurements, one with their eye open and the other with closed eye. The open eye readings measured both tear film and skin evaporation within the goggle, whereas the closed eye readings assessed skin evaporation only. Signals from the probe were recorded by computer software (Workbench 5.0, Strawberry Tree Inc., Sunnyvale, ON, Canada) for PC Windows to determine evaporation. Five readings were taken every second, the program recording a total of six hundred evaporation readings for two minutes. However, to allow evaporation values to stabilize, only the last 300 readings were used to calculate the final evaporation rate. As the calculated evaporation rate for the open eye included the evaporation of the surrounding skin within the goggle, a digital image of the open eye was therefore taken to calculate the eye area using Image J 1.34 computer software (National Institute of Health, Rockville, MD, USA). An Excel spreadsheet was used to calculate the evaporation rate using the formula below. Evaporation readings, eye area, temperature and humidity were entered into the spreadsheet and the final evaporation rates were calculated.
$$\left\{\frac{G\times O-\left\{\left(G-A\right)\times C\right\}}{A}\right\}\times \left\{\frac{1586.885\;\mathrm{Pa}}{Pmax-P}\right\}$$
where:

*A* = Area of the eye (mm^2^);

*G* = Inside area of the goggle (mm^2^);

*O* = Mean raw evaporation for an open eye (g/m^2^/h);

*C* = Mean raw evaporation rate for a closed eye (g/m^2^/h);

$\frac{1586.885\;\mathrm{Pa}}{Pmax-P}$ = Correction factor for humidity and temperature;

*P* = Partial pressure of water vapor;

*Pmax* = Saturated water vapor pressure at temperature (Pa).

### 2.7. Non-Invasive Tear Break-Up Time (NITBUT)

In the current study, an HIRCAL grid and Keeler Tearscope Plus were used to evaluate the stability of tear film [19,21]. The stability of the tear film was assessed using a fine grid insert that was projected onto the tear film using the Keeler Tearscope Plus. The Tearscope with fine grid insert was fixed to a slit lamp for the purpose of magnification of the reflected grid image. The local distortion or blurring of the fine grid image reflected from the tear film and ocular surface was observed. The time in seconds between the blink and the first occurrence of the fin grid image disturbance represents the non-invasive tear break-up time [22]. This time was observed and recorded three times, and then the mean value was calculated in order to estimate the non-invasive tear break-up time. The tear break-up time of 10 s provides a sensitivity and specificity of 82 and 86% (respectively) for the detection of dry eye [23]. Three measurements of the NITBUT were made, their mean was calculated and the result was recorded. In addition to evaluating the tear break-up time, A Keeler Tearscope Plus provided with non-illuminated biomicroscope was also used to assess the integrity and thickness of the lipid layer of the precorneal tear film [24]. The stability of the tear film was assessed using a fine grid insert that was projected onto the tear film. The subject was instructed to blink normally while the observer monitored the reflected image of the precorneal fine grid after it was focused on the tear film [19]. The Tearscope Plus that was equipped with cold-cathode lighting system was attached to the slit lamp. The subjects were instructed to maintain a straight gaze at the light source and blink normally. Then, focused clear interferometric patterns were obtained by carefully adjusting the focus of the system. The lipid layer thickness was then estimated using the classification of Guillon and Guillon grading system. This grading system classifies lipid layer thickness into six grades starting from open and closed meshwork patterns (13–50 nm) to color fringes and globular patterns (90–180 nm) [25]. 

### 2.8. Statistical Analysis

Collected data were statistically examined using PASW Statistics version 19. Firstly, a Kolmogorov–Smirnov test was applied to the data to examine the distribution of the data. Analysis of normally distributed data was carried out using a repeated measured ANOVA and Tukey’s post hoc test. Data not following normal distribution were compared using Friedman’s test and the post hoc Wilcoxon rank sum test. Pearson’s (normally distributed) and Spearman’s (not normally distributed) tests were used to examine the correlation between parameters.

## 3. Results

### 3.1. Lipid Layer Thickness

Statistical analysis of LLT measurements showed that the dry environment affected LLT significantly (*p* = 0.031). The median grade of LLT dropped from grade 3 (50–70 nm) at 40% RH to grade 2 (13–50 nm) at 5% RH. A significant increase in LLT was seen after both modes of treatment, with a median LLT grade of 3 when Emustil was used for both protection (*p* = 0.004) and relief (*p* = 0.016) compared with 5% RH with no drop being used. Pairwise statistical analysis showed no difference between the two treatments (Figure 2), and so the oil-in-water emulsion gives the same thicker LLT irrespective of whether it is used before or after exposure to dry conditions.

### 3.2. Tear Film Evaporation Rate

The tear film evaporation rate increased sharply at 5% RH (Figure 3). The mean evaporation rate went from 40.46 ± 11.80 g/m^2^/h at 40% RH to 88.47 ± 30.05 g/m^2^/h at 5% RH (*p =* 0.001). No significant difference was seen in tear evaporation following the instillation of Emustil at both protection (99.61 ± 33.56 g/m^2^/h) and relief (83.77 ± 20.37 g/m^2^/h).

### 3.3. Non-Invasive Tear Break-Up Time

Parametric tests showed that the mean NITBUT at 5% RH was significantly lower than that observed at 40% RH (Figure 4). Mean NITBUT dropped from 13.6 s at 40% RH to 6.6 s at 5% RH (*p* = 0.002). All NITBUT measurements at 5% RH (with or without the instillation of Emustil) were significantly lower than those at 40%. The instillation of Emustil at 5% RH resulted in a significant improvement in NITBUT for protection (*p* = 0.016) but this was not the case for relief (*p* = 0.0.56). The mean NITBUT increased to 8.5 s for protection and 8.0 s for relief.

## 4. Discussion

Dry eye disease (DED) is caused by several ailments and environmental dynamics. A variety of tear film supplements with a range of ingredients for managing the signs and symptoms of dry eye have been developed [26]. Although tear film supplements do not cure dry eye, they can help to reduce the signs and symptoms of dry eye and attempt to restore a normal homeostatic state at the ocular surface [27,28]. These products also play a role in substituting for tear fluid, reducing tear osmolarity and inhibiting the inflammatory process [26,29].

As tear film is aqueous-based, emulsions comprising both water and lipids have been developed as therapeutics [30]. Oil-in-water emulsions contain oily droplets preserved by surfactants or emulsifiers [31]. Most pharmaceutical emulsions contain submicron-sized particles prepared with oils (e.g., sesame oil, castor oil, soya oil, corn oil, glycerin monostearate, etc.) and emulsifiers (e.g., phospholipid, polysorbate 80, Cremophor^®^ RH, poloxamer 407, tyloxapol, etc.). Emulsions can be anionic or cationic based on the constituents added to the formulation during the emulsion process. Oil-in-water emulsions are well tolerated and often used as topical ocular drug transport carriers to improve membrane penetrability and the cellular uptake of lipophilic molecules [30,32].

As Emustil drops contain 3% natural phospholipids derived from egg yolk, Korb et al. demonstrated that the positively charged molecules in the egg yolk lecithin molecule would function as a potential promotor for oil-in-water emulsions. Oil-in-water emulsions have been proven to be very beneficial, such as phosphatidyl glycerol, phosphatidyl, inositol and phosphatidyl serine. This is due to their negative charges, which are extremely beneficial to oil-in-water emulsions [33]. A previous study indicated that Emustil effectively enhanced tear volume and minimized corneal injury when instilled four times a day for a week either as a monotherapy or in a recipe with sodium hyaluronate in a rodent model of dry eye [34].

We aimed to evaluate the efficacy of Emustil in two different treatment modules in managing tear film parameters in subjects under dry environmental conditions. A panel of objective and subjective measures were carried out to observe tear film parameters at normal (40% RH) and dry (5% RH) environmental conditions. An improvement in the signs and symptoms of dry eye has been previously reported following the use of a different oil emulsion (castor oil) [35]. In the current study, the Emustil drop was instilled both before (protection) and after the exposure (relief) to the desiccating environment to determine the optimum phase to use the therapy.

The human cornea is protected by an aqueous tear film coated by a lipid outer layer called the tear film lipid layer (TFLL). The TFLL moderates the surface tension of the tear film and comforts the film’s re-spreading after blinks. Fluctuations in tear lipid composition and properties are associated with DED. The tear film is not a stationary structure because of three fundamental processes: tear flow, evaporation and blinking. While the dynamics linked with tear flow and evaporation can lead to a stationary process, blinking is a cause of substantial non-systematic instabilities of the tear film [36]. We noticed a dramatic reduction in tear film LLT in normal healthy subjects after exposure to conditions of stress. The median LLT dropped from grade 3 (50–70 nm) at normal humidity to grade 2 (13–50 nm) in dry conditions. The tear lipid layer distortion could result from the disturbance in the structure of the polar lipids layer at a low humidity [37,38,39]. The increase in tear lipid layer thickness could be due to the improvement in aqueous layer thickness or the lipid spreading rather than the enhancement in meibomian gland secretion [16,40].

In this study, a thicker lipid pattern was seen after the administration of an Emustil eye drop in both the protection and relief method, with a median grade of 3. A very thick irregular lipid pattern (color fringes—grade 5) was also seen after the instillation of a drop in protection and relief. It should be noted that the color fringes pattern can be considered as an abnormal lipid pattern with a poor spreading ability [21]. The LLT was assessed immediately post instillation of the tear supplement in the relief mode and after 15 min in protection. Therefore, the presence of this irregular thick pattern could result from the sudden increase in LLT, which may disappear after some time passes. The appearance of color fringes is in agreement with previous studies, which reported that a thick lipid layer of 120 and 146 nm (color fringes pattern) had been found following the instillation of a single drop of mineral oil (Soothe eye drop) [17,41]. This increase in lipid layer thickness could be because the emulsion drop is made from natural lipids and is designed to improve the tear lipid structure and function. In addition to a high content of phospholipids (30%), Emustil has a ratio of non-polar/polar lipids of 0.67:1, which is very similar to that of natural meibomian lipids [17]. This composition could improve Emustil’s ability to supplement the lipid layer of the tear film [17] and thus increase lipid thickness during exposure to dry environments.

The evaporation rate of the tear film is mainly controlled by the lipid layer [42]. The loss of anti-evaporative properties of the lipid layer is frequently associated with dry eye conditions [43]. Earlier analyses have revealed that reduced lipid secretion results in a higher tear evaporation rate. Tear film thinning after blinks is usually attributed to the evaporation of water from the aqueous subphase [44,45]. A minimal reduction in tear evaporation was observed in relief; however, statistical tests showed that the reduction in tear evaporation was not significant. Our findings contradict previous research that reported a significant reduction in the tear film evaporation rate following the use of oil-in-water emulsion eye drops. There are several potential explanations for this discrepancy [15,16]. Emustil formulation was found to thicken the tear lipid layer. However, it should be noted that an abnormal irregular thick lipid pattern (color fringes) was found on five occasions (three in relief and two in protection) following the instillation of Emustil. This is supported by the fact that the mean evaporation rate seen in the individuals when the color fringes pattern was present was 101 g/m^2^/h. The mean evaporation of all individuals who did not show color fringes was 86.53 g/m^2^/h. Thus, although the instillation of Emustil eye drops increased the thickness of the lipid layer, this sudden thickening could result in irregularity and a poor spreading of the lipid layer, which may affect the lipid layers’ ability to inhibit tear evaporation.

The amount of tears in the eye is reliant on two dynamics, drainage through the lacrimal passages and evaporation. Issues like a declined tear production, prompt evaporation rate, instability of the tear film, tear hyperosmolarity, inflammations, ocular surface damages, etc., can contribute to dryness of the eyes [46]. The tear break-up time (TBUT), also known as the tear film break-up time (TFBUT), is the time consumed for the initial dry spot to appear on the cornea after a complete blink. TFBUT is a rapid technique used to evaluate the stability of the tear film. It is a routine investigative method in dry eye clinics [47]. In the non-invasive TFBUT measurement method, a grid or concentric ring configuration is projected onto the cornea and the subject is requested to blink. If the cornea is dry, the ring will appear as distorted. The time interval amid the last blink and the alteration of the ring configuration provides the measurement of the non-invasive tear break-up time (NITBUT) [48]. In the current investigation, the formation of a thick irregular lipid layer post instillation of Emustil is supported by NITBUT results. Pairwise statistical analysis indicated that, when used for protection, a significant improvement in NITBUT was seen, whereas, in relief, there was no significant improvement. This may indicate that the instillation of Emustil before exposure to adverse conditions allows the lipid layer to reform its regular and smooth spread over the aqueous layer, providing stability to the tear film. However, the difference in NITBUT between protection and relief could also be because the measurement was taken immediately after instillation in relief but after 15 min for protection. The improvement in tear stability observed when using the oil-in-water emulsion agrees well with the result of previous studies [17,49,50]. This finding can be expected as the oil–water emulsion is designed to enhance the lipid layer structure and function to enhance the stability of the tear film.

Sample size power calculation suggested that a study on a total of 28 subjects would detect a statically significant difference in tear osmolarity between 40% and 5% RH (at *p* = 0.05, two-sided, power 0.8) for the test population. Therefore, further studies on a larger sample size may be needed to confirm these results. An alteration in tear film parameters such as the evaporation rate, production and stability of tears in an adverse dry environment could affect the normal homeostatic condition of the ocular surface. Therefore, a compensatory mechanism is expected to occur to restore the normal homeostatic status [51]. This process includes increased blinking and the stimulation of reflex secretion from the lacrimal and meibomian glands [51]. This study suggests that using Emustil oil-in-water emulsion either before or post exposure to a dry environment could help people who work in such environments.

## 5. Limitations of the Study

No females were involved in this study. During certain phases of the menstrual cycle, women may experience hormonal fluctuations that can affect the quality and quantity of tear film affecting its parameters. Given the potential impact of hormonal fluctuations on tear film stability, we chose to not include female subjects in this study to minimize the potential confusing effects of hormonal changes. This can help to ensure more consistent and reliable data and improve the ability to draw accurate conclusions from the study results.

## 6. Conclusions

This study shows that an oil-in-water emulsion eye drop is effective in the relief and protection of tear film parameters against an adverse dry environment. A single instillation of Emustil was shown to improve LLT and stability in ultra-dry conditions (5% RH). Tear film stability was markedly better in the protection technique. This may be due to the sudden increase in lipid layer thickness resulting in a thick irregular lipid surface with a poor spreadability as seen by the frequency of color fringes patterns observed when the emulsion was used post-exposure for the relief of symptoms. However, both treatment modalities could be affected for restoring normal tear film parameters under desiccating environmental conditions. Although, in the current investigation, the sample size was relatively small, many tear film parameters were evaluated. Human tear film parameters, including the thickness of the lipid layer (LLT), evaporation rate and tear film stability, were evaluated at different time points. Despite the small sample size potentially limiting the ability to generalize the findings to a larger population, the result of the current study found significant changes in the tear film. However, in future investigations, it would be useful to evaluate the effect of the tear supplement with a larger sample size, which may increase the statistical power of the analysis.

## Figures and Tables

**Figure 1 medicina-59-01298-f001:**
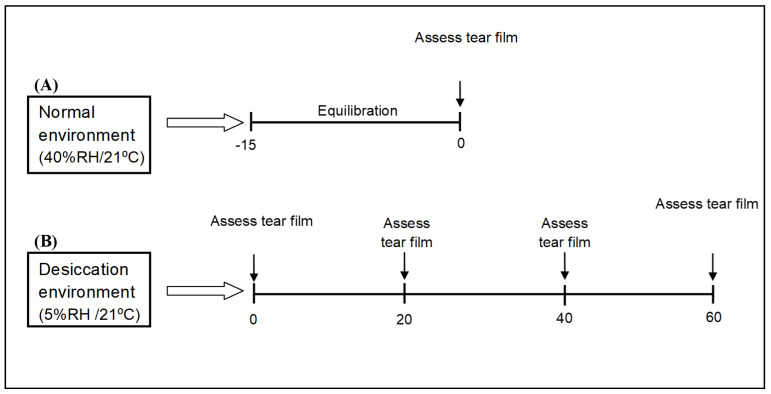
Study design. (**A**) Tear film parameters were assessed in normal environmental condition. (**B**) Tear film parameter investigations were carried out at different time points (immediately, and then at 20, 40 and 60 min) during the exposure to dissociating environment.

**Figure 2 medicina-59-01298-f002:**
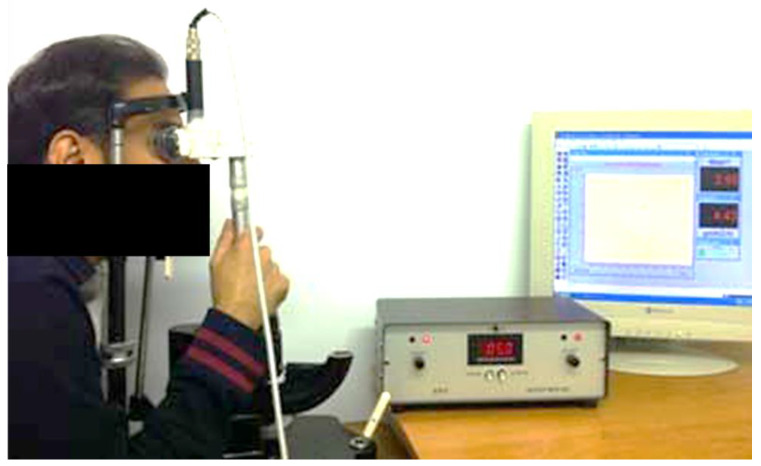
Measuring evaporation of tear film. The figure shows the probe attached to swimming goggles, Servo Med EP Evaporimeter and data collection software.

**Figure 3 medicina-59-01298-f003:**
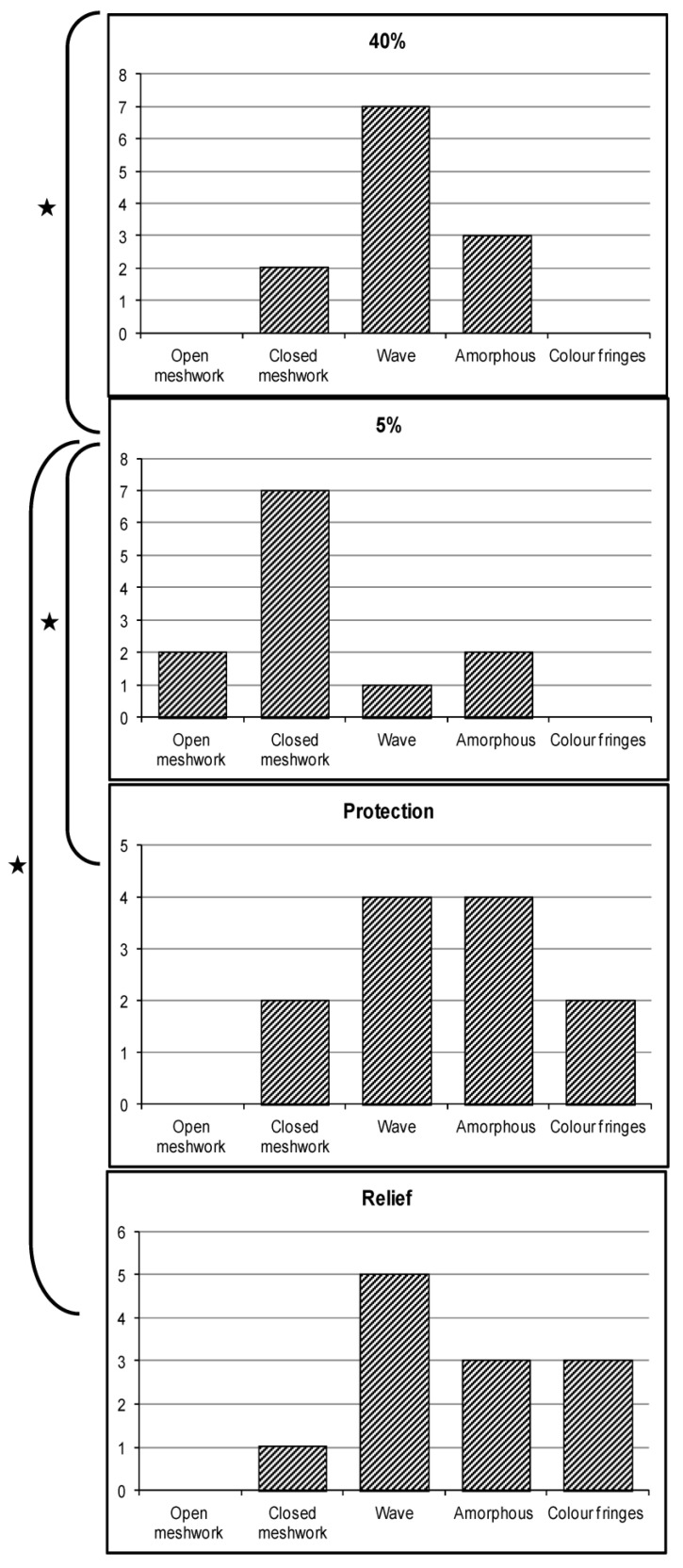
Histogram showing the frequency distribution of lipid patterns observed at normal and dry conditions and with the use of Emustil for protection and relief. Pairwise significant differences are indicated by (★).

**Figure 4 medicina-59-01298-f004:**
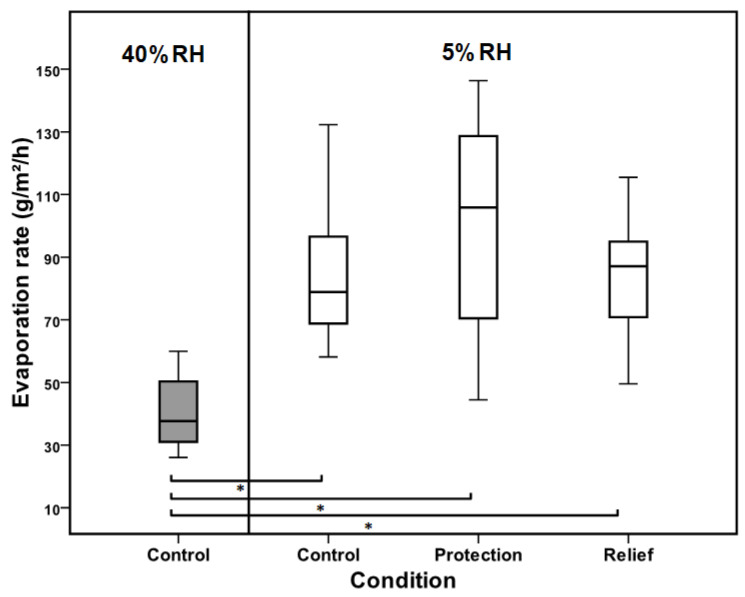
A box plot showing tear film evaporation measured at 40 and 5% RH without eye drop (control) and following the instillation of Emustil for the protection and relief methods (*n* = 12). Tear evaporation was significantly high at 5% and no improvement was seen with the use of Emustil. Pairwise significant (Tukey’s post hoc test) differences are indicated by (*).

## Data Availability

All the relevant data have been provided in the manuscript. Supplementary datasets used and/or analyzed during the current study are available from the corresponding author upon reasonable request.

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
