# Peer review of "Evaluation of Therapeutic Capability of Emustil Drops against Tear Film Complications under Dry Environmental Conditions in Healthy Individuals"

_medicina, 2023, doi:10.3390/medicina59071298_

Round 1

Reviewer 1 Report (Previous Reviewer 1)

The authors corrected the needed points by reviewers and manuscript is accepted to be published

Author Response

Thanks

Reviewer 2 Report (Previous Reviewer 2)

The authors have addressed my previous concerns. However, the introduction now has become way too long and impacts the readability of the manuscript. The authors should shortly refer to the 'big problem to be solved', weigh in on the knowledge gap, and end with concise aims. Intrinsic limitations such as a very small sample size remain (but I will not object to publication, since the results from multiple measurements appear to be consistent).

A native speaker or the language department of the university should perform a thorough check.

Author Response

The introduction section has been shortened

Language check has been done

Reviewer 3 Report (Previous Reviewer 3)

The authors have addressed my comments appropriately. Just a note that the limitation section should be before the Conclusion section.

Author Response

Limitation section has been incorporated before conclusion

This manuscript is a resubmission of an earlier submission. The following is a list of the peer review reports and author responses from that submission.

Round 1

Reviewer 1 Report

Then manuscript is well written , well organized the introduction is co incised , to the point, and covers the title and its aim, the ,methods are well organized but the number of cases is small 10 patients only and its well known that DED is more common in females so, it would be stronger, but does not affect the statistical results

Author Response

Dear reviewer

We have addressed the comments raised by you, please consider

Then manuscript is well written, well organized the introduction is co incised, to the point, and covers the title and its aim, the, methods are well organized but the number of cases is small 10 patients only and its well known that DED is more common in females so, it would be stronger, but does not affect the statistical results

Thank you for taking the time to review our work and provide your feedback. We appreciate your thoughtful comments and we agree with your observations and suggestions. Although the sample size is relatively small, many measurements have been conducted. In the current study number of tear film parameters have been evaluated. The human tear film parameters included lipid layer thickness (LLT), evaporation rate and tear breakup time were evaluated at different five points. Despite that the small sample size may limit the ability to generalize the findings to a larger population, the result of the current study found significant changes in tear film. However, in future investigation it would be useful to evaluate the effect of the tear supplement with a larger sample size that may to increase the statistical power of the analysis.

Reviewer 2 Report

In this study, Abusharha and colleagues investigate the effect of oil in water emulsion drops on the tear film of healthy subjects in dry environmental conditions. They report significant improvement in several tear film parameters when the emulsion drops are used for relief, but mostly when used for protection. The study is relatively clear and the statistical analysis is appropriate. Some comments to improve readability are provided below.

Abstract, line 20: Define RH.

Introduction, lines 75-81: This paragraph breaks the logical sequence of the introduction. Lines 75-78 should be moved to the beginning of the previous paragraph, while lines 78-81 should be moved inside the previous paragraph, before discussing studies in line 66.

Methods, lines 103-104: Which dry eye questionnaire was used?

Methods, lines 123-126: Details on the randomization are unclear. Please specify how many subjects were used for the relief and how many for the protection experiment. How many received treatment and how many did not? If the same patients were used for the two experiments, what was the order in which they were performed and what was the interval between the two experiments? Maybe a timeline graph would help.

Methods, lines 131: Which chapter are the authors referring to?

Methods: How many graders were used for the evaluation of tear film parameters?

Discussion, lines 218-243 and throughout: These lines better belong to the Introduction. Discussion should begin with a qualitative reiteration of all the findings. The authors should then discuss each finding separately and compare with literature.

Discussion, lines 300-301: It is difficult to claim that the effect was different, even though the P value of 0.056 is deemed as not significant, it is not very different from 0.016 substantially.

Discussion, lines 310-318: Discussion of study limitations is rather poor. Also, if the power analysis asked for more subjects, why did recruitment stop at n=12?

Conclusions, line 323: ‘markedly’ is an overestimation.

Language editing is necessary.

Author Response

Dear Editor

We are pleased to submit the revise version of the manuscript and addressed all the comments raised by reviewers

Reviewer  

In this study, Abusharha and colleagues investigate the effect of oil in water emulsion drops on the tear film of healthy subjects in dry environmental conditions. They report significant improvement in several tear film parameters when the emulsion drops are used for relief, but mostly when used for protection. The study is relatively clear and the statistical analysis is appropriate. Some comments to improve readability are provided below.

Thank you for taking the time to review our work and provide your feedback. We appreciate your thoughtful comments and we agree with your observations and suggestions.

Abstract, line 20: Define RH.

Done

Introduction, lines 75-81: This paragraph breaks the logical sequence of the introduction. Lines 75-78 should be moved to the beginning of the previous paragraph, while lines 78-81 should be moved inside the previous paragraph, before discussing studies in line 66.

Changes have been made according to the reviewer suggestions (lines 63 and 75)

Methods, lines 103-104: Which dry eye questionnaire was used?

(Ocular Surface Disease Index OSDI) of, it is added to the manuscript. (line 121)

Methods, lines 123-126: Details on the randomization are unclear. Please specify how many subjects were used for the relief and how many for the protection experiment. How many received treatment and how many did not? If the same patients were used for the two experiments, what was the order in which they were performed and what was the interval between the two experiments? Maybe a timeline graph would help.

Methods, lines 131: Which chapter are the authors referring to?

The sentence corrected (line 146).

Methods: How many graders were used for the evaluation of tear film parameters?

All study procedures and parameter assessment were carried out by one observer.

Discussion, lines 218-243 and throughout: These lines better belong to the Introduction. Discussion should begin with a qualitative reiteration of all the findings. The authors should then discuss each finding separately and compare with literature.

Thank you for your feedback, In response to your feedback, some unnecessary numbers are delated from the discussion part. In terms of relating the findings to other studies in the literature, we have conducted a review of the relevant literature and discussed previous studies that have examined the relationship between oil in water eye drops and evaporation rate (lines 68 to 75). However, we added to the discussion section to address contradictory findings in tear evaporation rate before explain the reasons. (lines 282 – 285)

Discussion, lines 300-301: It is difficult to claim that the effect was different, even though the P value of 0.056 is deemed as not significant, it is not very different from 0.016 substantially.

Thank you for your feedback and we have carefully considered your feedback. We agree with you that the result just miss the significant statistical significant. Therefore, we amended the conclusion part to clarify that both modalities could be helpful to restore normal tear film parameters under desiccating environmental conditions as there is no significant difference.  (lines 325 – 335)

Discussion, lines 310-318: Discussion of study limitations is rather poor. Also, if the power analysis asked for more subjects, why did recruitment stop at n=12?

appreciate your valuable comments and we agree with your observation. Although the sample size is relatively small, many measurements have been conducted. In the current study number of tear film parameters have been evaluated. The human tear film parameters included lipid layer thickness (LLT), evaporation rate and tear breakup time were evaluated at different five points. Despite that the small sample size may limit the ability to generalize the findings to a larger population, the result of the current study found significant changes in the tear film. However, in future investigations, it would be useful to evaluate the effect of the tear supplement with a larger sample size that may to increase the statistical power of the analysis.

Conclusions, line 323: ‘markedly’ is an overestimation.

Thank you for your feedback and we have carefully considered your feedback in revising the conclusion section.  We have made changes to the conclusion section as you suggest. We concluded in the revised conclusion that both modalities could be helpful to restore normal tear film parameters under desiccating environmental conditions.  (lines 325 – 335)

Reviewer 3 Report

This is a unique study which looks at the impact of instilling an oil-in-water emulsion eyedrop (Emustil) on dry eye disease parameters in a prophylactic/protective regimen or relief method. The Controlled Environment Chamber (CEC) also provides the authors the ability to specify and control the environmental conditions, which is an advantage particularly in evaluating dry eye disease therapeutics. While the findings are compelling, some concerns remain especially in regards to the presentation and terminologies used:

Abstract & Introduction

1. In the abstract line 20, define what the abbreviation RH is before first use.

2. The authors should briefly state the drop regimen for their protection and relief regimens in the abstract. This is one of the most important aspects of methodology, so this should be specified.

3. In the introduction lines 47-49, it sounds a bit confusing to say that 'Dry eye chronic cases... endure with potential damage to the ocular surface'. The authors should amend and reclarify this statement.

4. Line 58 of introduction: Be specific with terms; authors should state 'Eyedrop formulations', rather than 'Many formulations', given that no mentioning of eyedrops have been done in the introduction beforehand.

5. Line 59 of introduction: 'To better... increment the tear film' is a confusing term, this should be removed or amended.

6. Lines 67-69 of introduction: The authors mentioned Emustil improved signs and symptoms of dry eyes for Emustil in a study comparing 3 tear supplements; did the other 2 supplements also improve signs and/or symptoms? The authors should briefly mention this to give context.

7. The authors should make it very clear in their aims that they are comparing one regimen of treatment over the other, in its current form it sounds quite unclear. This should be clarified in the title as well, that the study is comparing a prophylactic versus relief method. Terms such as versus or compare against should be used to clearly show that there is a comparison being made. This was done well in the Methods section, as follows, but should be made much clearer earlier in the manuscript.

Methods

1. Lines 119-121: As per comment 7 above, the authors should specify this early in the abstract and introduction to set the scene. The terms 'pre- and post-exposure' are excellent terms to use, and should be specified earlier in relation to protection / relief when they are first mentioned.

2. A simple graphical timeline would contribute to the clarity of this study design, especially to compare the protection and relief methods. 

3. Line 131: Not sure what the Chapter 5 is referring to here, the authors should amend this. 

4. Lines 156-168: Why were two NITBUT techniques used? Could the authors explain the rationale behind this? 

Results

1. Figure 3: The authors should comment on whether these improvements are clinically meaningful even if it is statistically significant (especially when comparing 'Protection' against 'Control'. It seems like the improvement is miniscule. This should be explained in the Discussion. 

This would also impact the Conclusion (comment specific to this at the end of all these queries)

Discussion

1. The second paragraph of the Discussion should be inserted and/or integrated into the Introduction. This is the context, and readers need this context early to fully understand the thrust of the study. However, this is presented late in the manuscript, and should be moved from its current location.

2. Paragraph containing lines 268-286: The discussion should detail how the results could be explained and how the findings relate to other studies in the literature. It should not present new numerical findings.The new results in this paragraph should be included in the Results section.

3. Line 287: 'The amount of tear in the eye is liable on two dynamics', reliant may be a more accurate term than liable.

4. Lines 288-290: '... can source dryness to the eyes' should be rephrased. The authors could use '... can contribute to dryness of the eyes' instead.

5. Lines 299-301: ' p-values are unnecessary here given the authors have already stated these in the results section.

6. No females were involved in this study. This should be mentioned as a limitation as well in terms of the generalisability of this study.

Conclusion

The authors should be quite conservative with their conclusion. Lipid layer thickness improved with both 'protection' and 'relief' regimens, but the authors have stated 'pairwise statistical analysis showed no difference between the two treatment' in terms of lipid layer thickness. NITBUT improved only in 'protection' regimen. And evaporation rate was unchanged. 

In the conclusion, the authors mentioned that 'Tear film stability was markedly better in protection technique.', although the findings and graphs show miniscule clinical improvement, and statistical analysis showed no significant difference between protection and relief methods for NITBUT improvement. Hence, it may be misleading to state that protection is 'markedly better', when the evidence suggests that the improvement only in NITBUT compared to controls may only be marginal, with statistical analyses of all the parameters showing no major difference between protection and relief methods. The authors should amend their conclusion to reflect this.

Spelling and punctuation should be checked throughout the manuscript.

Author Response

Dear Reviewer 

we have addressed all the comments raised by you, please consider

Review2

This is a unique study which looks at the impact of instilling an oil-in-water emulsion eyedrop (Emustil) on dry eye disease parameters in a prophylactic/protective regimen or relief method. The Controlled Environment Chamber (CEC) also provides the authors the ability to specify and control the environmental conditions, which is an advantage particularly in evaluating dry eye disease therapeutics. While the findings are compelling, some concerns remain especially in regards to the presentation and terminologies used:

Thank you for taking the time to review our manuscript. We appreciate your feedback and have carefully considered your comments. We agree that further clarification was needed in certain areas of the manuscript, and we have taken steps to address this.

Abstract & Introduction

  1. In the abstract line 20, define what the abbreviation RH is before first use.

Response: Done

  1. The authors should briefly state the drop regimen for their protection and relief regimens in the abstract. This is one of the most important aspects of methodology, so this should be specified.

Response: More details about treatment modalities used in current current study have been added to the abstract to ensure that the methods are clearly and accurately presented in the abstract. (lines 20 to 22)

  1. In the introduction lines 47-49, it sounds a bit confusing to say that 'Dry eye chronic cases... endure with potential damage to the ocular surface'. The authors should amend and reclarify this statement.

Response: Sentence has been amended (lines 51 to 54)

  1. Line 58 of introduction: Be specific with terms; authors should state 'Eyedrop formulations', rather than 'Many formulations', given that no mentioning of eyedrops have been done in the introduction beforehand.

Response: We agree with the reviewer, the term has been changed.

  1. Line 59 of introduction: 'To better... increment the tear film' is a confusing term, this should be removed or amended.

Response: The term is removed.

  1. Lines 67-69 of introduction: The authors mentioned Emustil improved signs and symptoms of dry eyes for Emustil in a study comparing 3 tear supplements; did the other 2 supplements also improve signs and/or symptoms? The authors should briefly mention this to give context.

Response: Yes, the other two supplements did improve the tear parameters. The sentence is amended and the other 2 drops is mentioned in the introduction. (lines 72 – 75)

  1. The authors should make it very clear in their aims that they are comparing one regimen of treatment over the other, in its current form it sounds quite unclear. This should be clarified in the title as well, that the study is comparing a prophylactic versus relief method. Terms such as versus or compare against should be used to clearly show that there is a comparison being made. This was done well in the Methods section, as follows, but should be made much clearer earlier in the manuscript.

 Response: We appreciate your thoughtful comments and we agree with your suggestion. Comparing treatment modality is now added to the manuscript earlier and stated clearly in the aim. ( lines 97 – 99 ).

Methods

  1. Lines 119-121: As per comment 7 above, the authors should specify this early in the abstract and introduction to set the scene. The terms 'pre- and post-exposure' are excellent terms to use, and should be specified earlier in relation to protection / relief when they are first mentioned.

Response: We agree with the reviewer. The terms post and pre exposure are added earlier in the manuscript ( lines 95 – 96)

  1. A simple graphical timeline would contribute to the clarity of this study design, especially to compare the protection and relief methods. 

Response: Figure 1 is added to the manuscript to illustrate the study design.

  1. Lines 156-168: Why were two NITBUT techniques used? Could the authors explain the rationale behind this? 

Response: We illustrate that a Keeler Tearscope Plus was used during the main study (see line 175 please). the HIRCAL grid method was used only during screening visit for the purpose of either include or exclude the subjects.

Results

  1. Figure 3: The authors should comment on whether these improvements are clinically meaningful even if it is statistically significant (especially when comparing 'Protection' against 'Control'. It seems like the improvement is miniscule. This should be explained in the Discussion. 

This would also impact the Conclusion (comment specific to this at the end of all these queries)

Response: In this study, we observed statistically significant improvements in the outcome measures. However, we agree with the reviewer it is essential to consider whether these improvements are not only statistically significant but also clinically meaningful. Although the observed effect sizes may appear minuscule, it is important to assess their clinical relevance in the context of the specific condition and treatment being studied. In some cases, even small improvements can have a meaningful impact on patient’s health and life quality. For instance, a modest increase in tear break up time for only two or three seconds might have important implications for patients at ocular surface integrity and ocular discomfort symptoms that could translate into better quality of life for patients with chronic conditions such as dry eye.

Discussion

  1. The second paragraph of the Discussion should be inserted and/or integrated into the Introduction. This is the context, and readers need this context early to fully understand the thrust of the study. However, this is presented late in the manuscript, and should be moved from its current location.

Response: The paragraph is moved to the introduction section (lines 68 – 71) .

  1. Paragraph containing lines 268-286: The discussion should detail how the results could be explained and how the findings relate to other studies in the literature. It should not present new numerical findings. The new results in this paragraph should be included in the Results section.

Response: Thank you for your feedback regarding the discussion section. We appreciate your suggestion to provide more detail on how the results could be explained and how they relate to other studies in the literature. In response to your feedback, we have revised the discussion section to provide a more detailed explanation of the key findings. Regarding the explanation of the results, we have taken care to consider the theoretical framework of   the study and relate the findings back to the research question and hypotheses. We focused on explaining why although we found an improvement in lipid layer thickness, the tear evaporation was not improved.  We agree that discussion should not present new numerical findings. Some unnecessary numbers are delated from the discussion part (lines 283 to 26). However, the other numbers presented here are just to illustrate that an abnormal irregular thick lipid pattern (colour fringes) could be the reason behind the high evaporation rate found in this study after using the oil eye drops. The numbers presented here are for these cases only (irregular thick lipid pattern) to provide more explanation for evaporation rate results rather than reporting  new results. In terms of relating the findings to other studies in the literature, we have conducted a review of the relevant literature and discussed previous studies that have examined the relationship between oil in water eye drops and evaporation rate (lines 68 to 75). However, we added to the discussion section to address contradictory findings in tear evaporation rate before explain the reasons. (lines 282 – 285)

  1. Line 287: 'The amount of tear in the eye is liable on two dynamics', reliant may be a more accurate term than liable.

Response: Phrase is changed to “reliant”.

  1. Lines 288-290: '... can source dryness to the eyes' should be rephrased. The authors could use '... can contribute to dryness of the eyes' instead.

Response: Sentence is changed as suggested.

  1. Lines 299-301: ' p-values are unnecessary here given the authors have already stated these in the results section.

Response: p-values are removed as suggested

  1. No females were involved in this study. This should be mentioned as a limitation as well in terms of the generalisability of this study.

Response: During certain phases of the menstrual cycle, women may experience hormonal fluctuations that can affect the quality and quantity of tear film affecting its parameters. Given the potential impact of hormonal fluctuations on tear film stability, we choose to not include female subjects from in this study in order to minimize the potential confusing effects of hormonal changes. This can help to ensure more consistent and reliable data and improve the ability to draw accurate conclusions from the study results.

Conclusion

The authors should be quite conservative with their conclusion. Lipid layer thickness improved with both 'protection' and 'relief' regimens, but the authors have stated 'pairwise statistical analysis showed no difference between the two treatment' in terms of lipid layer thickness. NITBUT improved only in 'protection' regimen. And evaporation rate was unchanged. 

In the conclusion, the authors mentioned that 'Tear film stability was markedly better in protection technique.', although the findings and graphs show miniscule clinical improvement, and statistical analysis showed no significant difference between protection and relief methods for NITBUT improvement. Hence, it may be misleading to state that protection is 'markedly better', when the evidence suggests that the improvement only in NITBUT compared to controls may only be marginal, with statistical analyses of all the parameters showing no major difference between protection and relief methods. The authors should amend their conclusion to reflect this.

Response: Thank you for your feedback and we have carefully considered your feedback in revising the conclusion section.  We have made changes to the conclusion section as you suggest. We concluded in revised conclusion that both modalities could be helpful to restore normal tear film parameters under desiccating environmental conditions.  (lines 325 – 335)

Round 2

Reviewer 3 Report

I appreciate the authors responses to my comments and suggestions. In regards to this comment no. 6 of the Discussion and the author's response, the authors should clearly specify that they considered females as an exclusion criteria, given that this is not clear in the report. The rationale behind this should also be clearly stated, either in the discussion or exclusion criteria of the methods.

Just as a note, excluding females based just on the reasoning that hormonal changes may impact tear quality may be detrimental to generalisability of findings, as this would exclude them from being recruited in dry eye disease studies, and translatability of findings to the overall population (which includes females) could be compromised. Hence, the authors should also still include this as a limitation of the study in addition to stating that females were excluded.

N/A

Author Response

Dear Reviewer

Thanks for highlighting the point 

We added in point in limitation of the study

Limitation of the study

No females were involved in this study. During certain phases of the menstrual cycle, women may experience hormonal fluctuations that can affect the quality and quantity of tear film affecting its parameters. Given the potential impact of hormonal fluctuations on tear film stability, we choose to not include female subjects from in this study in order to minimize the potential confusing effects of hormonal changes. This can help to ensure more consistent and reliable data and improve the ability to draw accurate conclusions from the study results.

Regards